Corrected: Author correction

# Insights into a dual function amide oxidase/ macrocyclase from lankacidin biosynthesis

Jonathan Dorival[1,4], Fanny Risser[1], Christophe Jacob[1], Sabrina Collin[1], Gerald Dräger [2], Cédric Paris[3], Benjamin Chagot [1], Andreas Kirschning[2], Arnaud Gruez[1] & Kira J. Weissman [1]

Acquisition of new catalytic activity is a relatively rare evolutionary event. A striking example appears in the pathway to the antibiotic lankacidin, as a monoamine oxidase (MAO) family member, LkcE, catalyzes both an unusual amide oxidation, and a subsequent intramolecular Mannich reaction to form the polyketide macrocycle. We report evidence here for the molecular basis for this dual activity. The reaction sequence involves several essential active site residues and a conformational change likely comprising an interdomain hinge movement. These features, which have not previously been described in the MAO family, both depend on a unique dimerization mode relative to all structurally characterized members. Taken together, these data add weight to the idea that designing new multifunctional enzymes may require changes in both architecture and catalytic machinery. Encouragingly, however, our data also show LkcE to bind alternative substrates, supporting its potential utility as a general cyclization catalyst in synthetic biology.

[1] UMR 7365, Ingénierie Moléculaire et Physiopathologie Articulaire (IMoPA), CNRS-Université de Lorraine, Biopôle de l'Université de Lorraine, Campus Biologie Santé, 9 Avenue de la Forêt de Haye, BP 20199, 54505 Vandœuvre-lès-Nancy Cedex, France. [2] Institut für Organische Chemie, Leibniz Universität Hannover, Schneiderberg 1B, Hannover 30167, Germany. [3] Laboratoire d'Ingénierie des Biomolécules, Ecole Nationale Supérieure d'Agronomie et des Industries Alimentaires (ENSAIA), Université de Lorraine, 2 Avenue de la Fôret de Haye, BP 172, 54518 Vandœuvre-lès-Nancy Cedex, France. [4] Present address: Sorbonne Universités, UPMC Univ. Paris 06, CNRS, UMR 8227, Integrative Biology of Marine Models, Station Biologique de Roscoff, CS 90074 Roscoff, Bretagne, France. These authors contributed equally: Fanny Risser, Christophe Jacob. Correspondence and requests for materials should be addressed to A.G. (email: arnaud.gruez@univ-lorraine.fr) or to K.J.W. (email: kira.weissman@univ-lorraine.fr)

Enzyme activity within superfamilies frequently evolves via changes in substrate specificity, whereas reaction chemistry is usually retained[1,2]. There are also well-documented examples of a conserved protein fold housing diverse catalytic activities: the αβ hydrophobic barrel fold accommodates at least six different types of acid–base chemistry[3], the fumarylacetoacetate hydrolase superfamily includes decarboxylases, isomerases, hydratases, and hydrolases[4], whereas the enolase superfamily has > 30,000 members, which share an interdomain active site architecture and the mechanistic strategy of proton abstraction α to a carboxylate and stabilization of the enolate by a magnesium ion[5]. This evolution of novel activity can occur via several different mechanisms, including the insertion or deletion of residues, changes in oligomerization state, and the recruitment of metal ions and cofactors[1]. The enzyme LkcE from the lankacidin polyketide biosynthetic pathway of the bacterium *Streptomyces rochei* represents an interesting test case for understanding such gain-of-function, as it exhibits a monoamine oxidase (MAO) protein fold yet also carries out a macrocyclization reaction.

The core of the lankacidin molecule is constructed from simple acyl-CoA and amino-acid-building blocks by a series of gigantic multimodular polypeptides, which operate in assembly-line fashion (polyketide synthases (PKSs) and non-ribosomal peptide synthetases (NRPSs), respectively)[6]. The full-length chain is released from the last multienzyme LkcG by a dedicated thioesterase domain[7]. Unusually for this type of pathway, the product is liberated from the multienzyme not as a macrolide, but as a β-oxo-δ-lactone (Fig. 1). LkcE then catalyzes post-assembly macrocyclization via formation of a C-2−C-18 carbon-carbon bond[8]. This reaction is proposed to occur on a C-7 acetylated metabolite LC-KA05 (**1**), although the timing of the acetylation reaction, and the enzyme responsible, are unknown. A strain of *S. rochei* in which LkcE is inactivated accumulates LC-KA05 (Fig. 1), which can be converted into lankacidin C (**3**) by a strain blocked early in lankacidin biosynthesis, demonstrating the chemical competence of LC-KA05 as an intermediate[8]. In the final stages of the biosynthesis, the C-24 hydroxyl group is oxidized to a ketone and the C-7 acetyl group is cleaved, yielding lankacidin C, with the

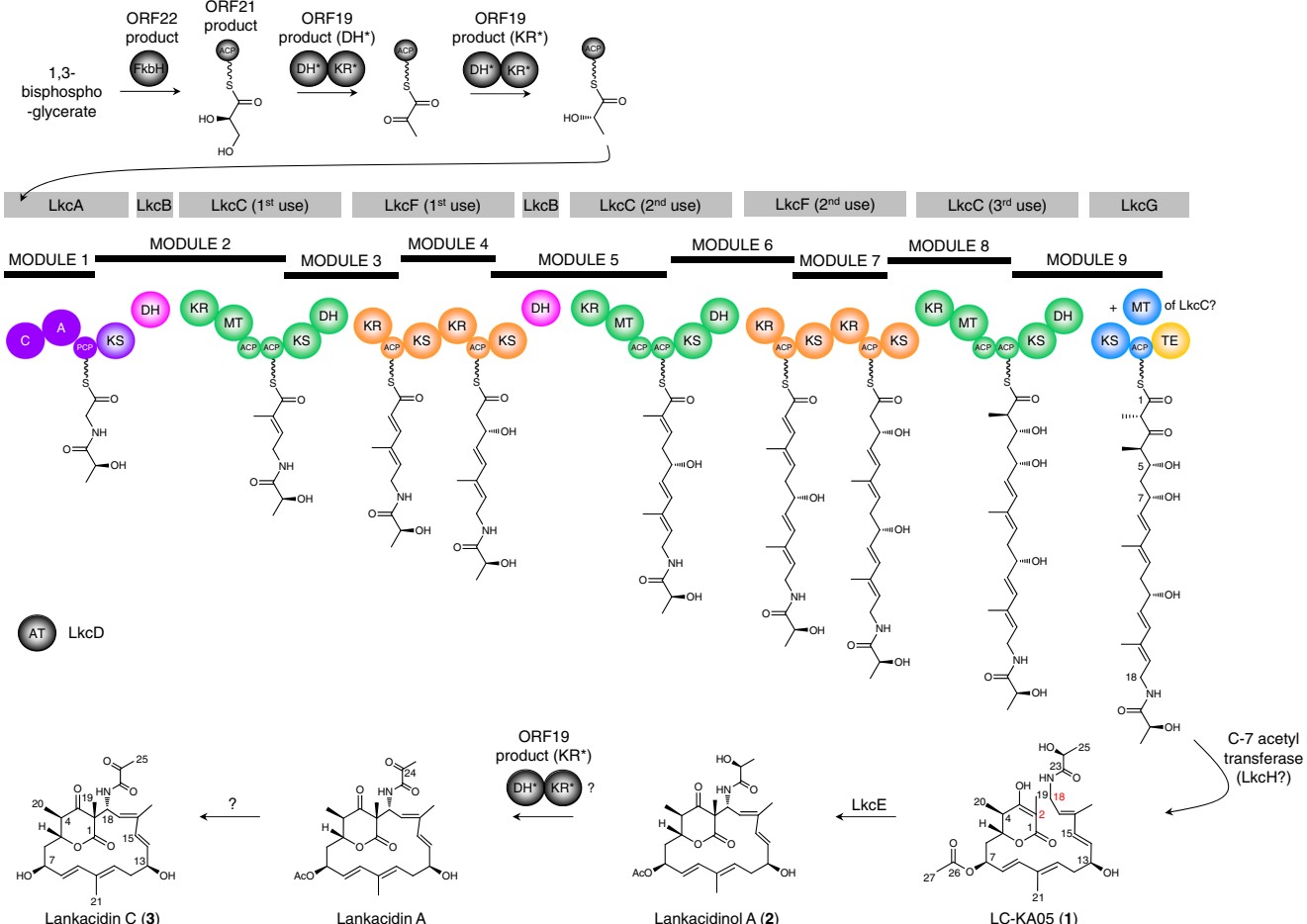

**Fig. 1** Proposed biosynthetic pathway to lankacidin C. The gene cluster, which has been shown to be sufficient for lankacidin biosynthesis[44], encodes five assembly-line proteins, LkcA (a hybrid NRPS/PKS subunit), LkcB (a discrete DH), LkcC, LkcF, and LkcG, together containing a total of only four KS domains, although eight KS-mediated extension cycles are required. One possibility that agrees with phylogenetic analysis of KS substrate specificity[23] is shown here, in which the assembly-line incorporates multiple copies of the proteins LkcB, LkcC and LkcF (see ref. [44] for an alternate view). The starter unit may be either pyruvoyl-ACP or lactoyl-ACP, both derived from 1,3-bisphosphoglycerate[45]. However, we and others[8] have found that in an LkcE-deleted mutant, only the lactoyl form of the first free intermediate LC-KA05 accumulates (Supplementary Fig. 8), so we propose that lactoyl-ACP serves as the starter unit. The enzyme responsible for acetyl transfer at C-7 likewise remains to be determined, although a candidate is LkcH, which shows homology to isochorismatases[8]. The object of this study, LkcE, catalyzes the critical macrocyclization reaction of LC-KA05 (**1**), resulting in lankacidinol A (**2**). The carbons implicated in the ring closure are indicated in red. Domain abbreviations: C, condensation; A, adenylation; PCP, peptidyl carrier protein; KS, ketosynthase; DH, dehydratase; KR, ketoreductase; MT, *C*-methyltransferase; ACP, acyl carrier protein; TE, thioesterase

**Fig. 2** Structures of selected compounds investigated in this study. Compound LC-KA05 (**1**) was shown by NMR to exist almost exclusively in the enol form, and thus its derivatives **6** and **7** are also represented as enols. The gray boxes indicate where **6** and **7** differ from **1**. The stereochemistry of the C-6−C-7 double bond in **7** and cyclized 7 (**8**) is unknown, as indicated

relative timing of these two transformations also remaining to be established (Fig. 1).

The catalytic mechanism proposed for LkcE involves initial oxidation of the amide function of LC-KA05 to an iminium ion[8]. This is followed by attack of a C-2 enolate anion on the iminium to give the 17-membered lankacidinol A (**2**)[8]—an enzymatic intramolecular Mannich reaction. Although numerous flavin adenine dinucleotide (FAD)-dependent amine oxidases have been characterized to date[9], to our knowledge LkcE is only the second described amide oxidase[10]. This observation, coupled with the fact that no other known enzyme carries out Mannich chemistry except as a promiscuous activity[11], motivated our interest in establishing a detailed structure/function relationship for LkcE.

In this study, we have used a combination of X-ray crystallography, structure-guided mutagenesis, and kinetic analysis, to reveal key architectural differences between LkcE and other members of the MAO family, which underpin its second, cyclization activity. We also demonstrate the tolerance of LkcE to certain structural modifications of the substrate. As 1,3-diketones and amide functional groups are common in PK/NRP hybrid metabolites, LkcE represents a potentially valuable addition to a synthetic biology toolbox as a means to access novel macrocyclic structures of various sizes and functionality.

## Results

**Structural characterization of holo and ligand-bound LkcE**. *S. rochei* LkcE was purified to homogeneity as a recombinant protein (Supplementary Tables 1 and 2) from *Escherichia coli* and its redox cofactor identified as FAD by mass spectrometry following chromatographic separation under denaturing conditions (Supplementary Fig. 1); thus in common with certain MAO family members[12] but distinct from the only other reported amide oxidase Af12070 (ref. [10]), the FAD cofactor is non-covalently bound. By UV-Vis, 45% of the protein was estimated to contain FAD (on a par with AF12070 (ref. [10])) (Methods). We crystallized the enzyme in its holo form, but additionally in the presence of substrate analogs: ethyl 2-methylacetoacetate (EMAA) (**4**) that mimics the δ-lactone of LC-KA05, and *N,N′*-diallyl-L-tartardiamide (DATD) (**5**), which resembles the amide region (Fig. 2). The X-ray crystal structure of the holo protein was solved at 3.15 Å resolution by single-wavelength anomalous dispersion (SAD) using data collected on seleniated protein (Fig. 3a). The structure

of LkcE (Fig. 3b, c) co-crystallized in the presence of the analogs was then solved at 2.80 Å by molecular replacement (the statistics for data collection, refinement, and validation of both structures are presented in Supplementary Table 3). For all four structures, > 99.7% of the residues were in allowed regions of the Ramachandran plot. In all but the SeLkcE structure, one Gly (299 or 300) was an outlier, whereas Glu313 was an outlier in both forms of the holo enzyme structure. However, for Glu313, clear electron density is present corresponding to the residue.

Sequence alignment (Supplementary Fig. 2) and comparison with the structures of the five closest structural homologs to LkcE identified by the DALI[13] server confirmed that LkcE belongs to the MAO family. The respective root mean square deviation (rmsd) of Cα positions was 4.232 Å (212 Cα) for 6-hydroxy-L-nicotine oxidase (6HDNO) from *Paenarthrobacter nicotinovorans* (PDB 3NG7)[14]; 4.720 Å (254 Cα) for human monoamine oxidase (hMAO) A (2Z5X)[15]; 5 Å (267 Cα) for hMAO B (1GOS)[16]; 10.239 (271 Cα) for protoporphyrinogen oxidase (PPOX) of *Bacillus subtilis* (3I6D)[17]; and 14.660 Å (162 Cα) for a polyunsaturated fatty acid isomerase of *Propionibacterium acnes* (2BAB)[18].

The LkcE monomer contains two domains, a FAD-binding domain and a substrate-binding domain (Fig. 3a), which are joined by a substantial number of loop regions. The cofactor-binding domain (which includes residues 1−45 incorporating a characteristic FAD-binding motif (xhxhGxGxxGxxxhxxh (x)$_8$hxhE(D)[19]), 70−83, 206−279, and 368−438), is topologically similar to other proteins with the '3-layer (BBA) sandwich' fold in the CATH database[20], and comprises a central four-stranded antiparallel β-sheet flanked on one side by five α-helices, and on the other side by a second three stranded, antiparallel β-sheet and three α-helices. The FAD cofactor is present at 100% occupancy (Supplementary Fig. 3), and as in other members of the MAO family, it adopts an elongated conformation. Although the specific FAD-binding residues differ among LkcE and its closest structural homologs (Supplementary Fig. 2), the types of interactions are similar, with the exception of those to the isoalloxazine ring. In all of the structures except LkcE and PPOX, the isoalloxazine is flanked by two bulky aromatic residues (Tyr, and in 3NG7 only, a Tyr and a Trp). In LkcE, the equivalent residue positions are Gly364 and Leu398, respectively, whereas in PPOX, the analogous amino acids are M413 and G449. Thus, it

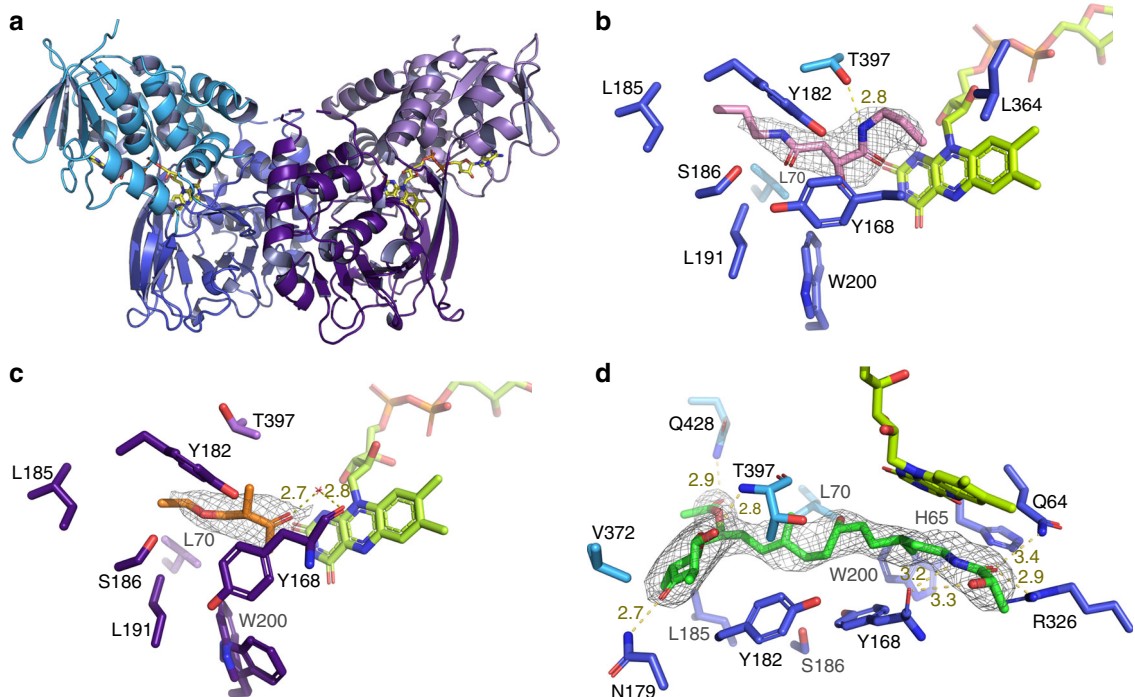

**Fig. 3** Crystal structures of LkcE and its mutants. **a** Crystal structure of homodimeric, wild-type LkcE. The FAD-binding domain is shown in light blue and light purple for the monomers A and B, respectively, whereas the substrate-binding domain is shown in dark blue and purple. The FAD is colored in yellow. **b** View of the active site of wild-type LkcE in the presence of bound DATD (pink) with its $2F_o - F_c$ map contoured at $1\sigma$. Binding occurs mainly via hydrophobic interactions and a single hydrogen bond with T397. **c** View of the active site of wild-type LkcE in the presence of bound EMAA (orange) with the $2F_o - F_c$ map contoured at $0.6\sigma$. Comparison with **b** shows the analog to be binding in the same region of the active site. **d** View of the active site of the LkcE E64Q mutant in the presence of bound LC-KA05 (green), which adopts a linear conformation. The $2F_o - F_c$ map surrounding the substrate is contoured at $1\sigma$

appears that the FAD-binding site has been modified in these two enzymes in order to accommodate the large macrocycles of the substrates/products.

The substrate-binding domain contains an orthogonal α-helix bundle that packs against a β-sheet. Visual inspection of the overlaid structures shows that although the cofactor-binding domain is well-conserved, the substrate-binding domain is significantly divergent in terms of both α-helical content and orientation (Supplementary Fig. 4). This is unsurprising given the pronounced structural differences between LC-KA05 and the typical small-molecule amine substrates of the MAO family[21].

Although the majority of characterized MAO family members are homodimeric, there is some disagreement between crystallographic and solution data, and at least one homolog is monomeric[9]. LkcE is clearly a homodimeric protein in the crystal, with an extensive interface (4705 Å$^2$, representing 15% of the total surface area). It is also homodimeric in solution, as shown by small-angle X-ray scattering (Fig. 4). However, it dimerizes differently to the human enzymes hMAO A[15] and B[16], as well as to its closest structural homolog 6HDNO[14] (Fig. 5), giving it a distinctive quaternary organization. In the hMAO B crystal structure, the dimerization interface is formed by both the cofactor- and substrate-binding domains, whereas in the case of 6HDNO, stabilization of the homodimer is additionally provided by two molecules of diacylglycerophospholipid. In contrast, the dimer interface of LkcE is formed exclusively by the substrate-binding domain of each monomer. The interface residues include E68, K69, V87, L106, F108, E114, D121, Q125, N128, Q129, S188, Y199, R201, H332, and R335. Most of these are well-conserved between LkcE and its nearest homologs, but not all (Supplementary Fig. 2). This distinct mode of dimerization, coupled with the

high content of loops at the interdomain interface within each monomer (Fig. 3a), would favor movement of the domains relative to each other, with potentially important implications for the catalytic mechanism, as discussed below.

**Structure-guided site-directed mutagenesis of LkcE.** Analysis of the crystal structures with substrate analogs bound revealed clear electron density attributable to EMAA and DATD, but only one of these compounds was present in any given active site, as their positions overlapped (Fig. 3b, c). Inspection of the two binding sites coupled with mechanistic considerations identified E64, H65, Y168, Y182, R326, and T397 as candidate catalytic residues (Supplementary Fig. 2). The presence of amino acids in the active site capable of acting as general acids/bases contrasts with other flavin-dependent amine oxidase homologs which lack such residues[14,21].

In particular, E64, Y182, and R326 are strictly conserved in all genes putatively encoding LkcE-type cyclases (Supplementary Fig. 2). Site-specific mutagenesis was used to produce five mutants: E64A, E64Q, Y182F, R326L, and R326Q, which were purified to homogeneity (FAD content, respectively: E64A, 58%; E64Q, 53%; Y182F, 42%; R326L, 20%; R326Q, 60%). Analysis by circular dichroism (CD) confirmed that the mutations had not dramatically altered the structures (Supplementary Fig. 5).

**Isolation of the native substrate of LkcE.** To obtain the native LkcE substrate LC-KA05, we disrupted gene *lkcE* in *S. rochei* by PCR targeting, using a strategy similar to that described previously[8,22,23] (Supplementary Figs. 6 and 7). Following analysis of the *lkcE* mutant by high-performance liquid chromatography-mass spectrometry (HPLC-MS) to verify the presence of LC-KA05

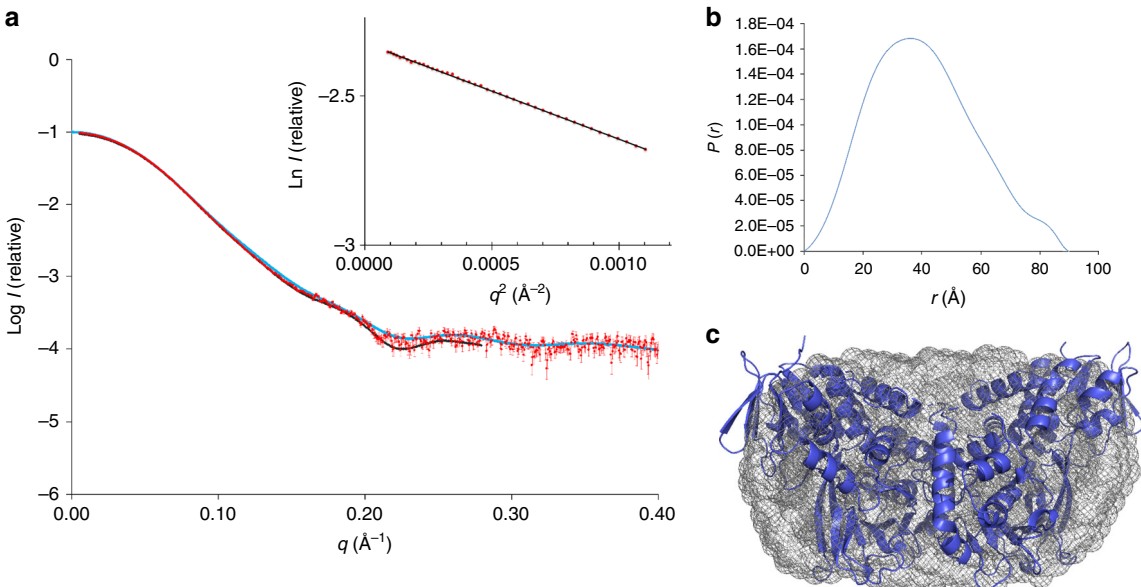

**Fig. 4** Characterization of recombinant LkcE by SAXS. **a** Fit between the *ab initio* model computed with GASBOR[46] (black line) ($\chi^2 = 1.44$), the theoretical scattering curve calculated on the basis of the structure of LkcE$_{WT}$ with CRYSOL[47] (blue line) ($\chi^2 = 3.8$) and the experimental SAXS data (red dots). Inset is the Guinier plot, which yields an $R_g$ of 31.0 Å. A molecular weight (MW) of 99.3 kDa was calculated using the SAXS MoW[48] program (homodimer calculated MW = 98.6 kDa). **b** The distance distribution function derived for LkcE calculated with GNOM[49], yielding a $D_{max}$ of 91 Å. **c** Averaged ab initio envelope of LkcE calculated using GASBOR[46] (gray mesh) with superimposition of the LkcE$_{WT}$ crystal structure carried out using SUPCOMB[49] (blue)

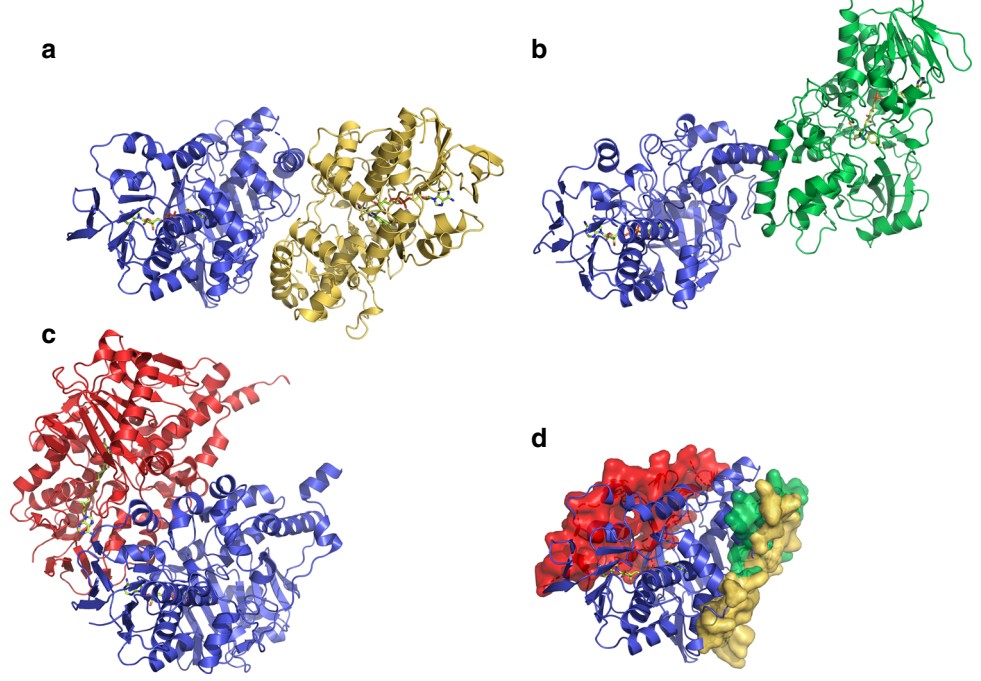

**Fig. 5** Alternate homodimerization mode of LkcE relative to hMAO B and 6HDNO. **a** Homodimeric structure of LkcE. **b** Homodimerization mode of 6HDNO. The dimer is superimposed on one monomer of LkcE (in blue in **a**), in the same orientation. **c** Homodimerization mode of hMAO B. Again, the dimer has been superimposed on a monomer of LkcE. **d** Interaction surface between the two monomers in LkcE (yellow), hMAO B (red), and 6HDNO (green), relative to a monomer of LkcE. This analysis clearly shows that the mode of homodimerization of LkcE, as well as its overall quarternary organization, differ from these two homologs

and the absence of all downstream metabolites (Supplementary Fig. 8 and Supplementary Table 4), **1** was obtained in pure form by preparative HPLC (Supplementary Fig. 9). Unfortunately, LC-KA05 was unstable, undergoing degradation via loss of the C-7 acetate, both by hydrolysis (to give 7-OH **6**, Fig. 2) and

elimination (to give **7** (Fig. 2)), which co-eluted with LC-KA05. However, 16 mg of mixed metabolite could be obtained from the extracts of 20−30 L of culture, with **1** and **7** present in an ~ 2:1 ratio (chemical data on all three compounds are provided in the Methods). As HPLC-MS analysis of extracts of both the wild-type strain

and the *lkcE* mutant did not reveal any evidence for **7**, nor, in the case of the wild type, for its cyclized equivalent **8** (Fig. 2; Supplementary Fig. 10 and Supplementary Table 4), the cellular environment must protect against this type of non-productive degradation. The 7-OH compound **6** was observed in extracts of the *lkcE* mutant although not in the wild type (Supplementary Fig. 10 and Supplementary Table 4), presumably because in the presence of active LkcE it can be productively cyclized to give lankacidinol C (**9**) (Fig. 2) (see below), and from there transformed to lankacidin C. Interestingly, NMR analysis of purified LC-KA05 revealed that the δ-lactone was almost exclusively in the enol form (Supplementary Fig. 9) as recently described for a new lankacidin-derived metabolite[24], presumably because the six-membered ring forces overlap in the π-system.

**Kinetic characterization of LkcE and its mutants**. Attempts to determine the kinetics of LkcE-catalyzed conversion of LC-KA05 to lankacidinol A by HPLC-MS were frustrated by the rapid degradation of **1** under the assay conditions (Supplementary Fig. 11). Nonetheless, larger scale incubation of **1** in the presence of LkcE produced authentic lankacidinol A, as judged by HPLC-MS comparison with extracts of wild-type *S. rochei* containing this metabolite, providing conclusive evidence that recombinant LkcE was active (Supplementary Fig. 12 and Supplementary Table 4). The enzyme was also found capable of cyclizing the degradation product **7**, as well as purified, deacetylated substrate **6**, to give lankacidinol C, as established by HPLC-MS (Supplementary Fig. 12 and Supplementary Table 4), evidence for its relaxed specificity toward structural variations at the C-6/C-7 ring positions.

As an alternative and more-sensitive kinetic method, we employed a coupled enzymatic assay to detect the $H_2O_2$ formed during each catalytic cycle. For this, we utilized NADH peroxidase from *Streptococcus faecalis*, which catalyzes the NADH-dependent reduction of $H_2O_2$ to water. In this way, LkcE turnover (production of $H_2O_2$) was detected via the consumption of NADH (loss of absorbance at 340 nm) under conditions where NADH peroxidase was not rate-limiting (see Methods)[25]. We also confirmed that the presence of dimethyl sulfoxide (used to solubilize the substrates) had no effect on the measured kinetic parameters (Fig. 6 and Table 1). However, as **1** was present at only ca. 66% in the mixture, and degraded spontaneously during the assays, the kinetic parameters must be considered lower estimates. Nonetheless, the possibility to rapidly acquire many data points in the initial stages of the reaction ($A_{340\,nm}$ was measured every 0.4 s) meant that effects of the decomposition of **1** were minimized.

With the ca. 2:1 LC-KA05/eliminated derivative **7** mixture, this system yielded the following steady-state kinetic parameters for the wild type (average of measurements in duplicate, and taking into account the observed proportion of holo LkcE (45%)): $k_{cat} = 3.4 \pm 0.2$ min$^{-1}$ and $K_M = 5 \pm 2$ μM. This $k_{cat}$ is comparable to both the overall rate of turnover reported for an intact PKS system in vitro (1.1 min$^{-1}$)[26], as well as that for the amide oxidase AF12070 (ca. 5 min$^{-1}$ (ref. [10])) (Fig. 6 and Table 1). Analysis of the wild-type enzyme with deacetylated derivative **6** yielded comparable parameters ($k_{cat} = 2.4 \pm 0.1$ min$^{-1}$, $K_M = 5 \pm 1$ μM) (Fig. 6 and Table 1). Mutants E64A, E64Q, and R326L were completely inactive with the **1**/**7** mixture (as was boiled protein control), whereas mutants Y182F and R326Q showed good activity towards the deacetylated derivative **6**: R326Q, $k_{cat} = 0.88$ min$^{-1}$, $K_M = 4$ μM; Y182F, $k_{cat} = 1.5$ min$^{-1}$, $K_M = 4$ μM (Fig. 6 and Table 1) (owing to limited quantities, the **1**/**7** mixture was not tested with these mutants). Taken together, these data are consistent with residues E64 and R326 playing

critical roles in the catalytic mechanism (although R can be substituted by Q without a significant effect on activity), whereas Y182 is not essential. We also confirmed lack of turnover with the substrate analogs EMAA and DATD, both separately and together (using 2 and 10 μM enzyme and up to 400 μM in both substrates), consistent with their overlapping binding modes (Fig. 3b, c).

**Crystal structures of LkcE mutants in complex with LC-KA05**. In order to obtain complexes with LC-KA05, mutants E64A, E64Q, Y182F, R326L, and R326Q were soaked with the **1**/**7** substrate mixture followed by rapid acquisition of the X-ray diffraction data (owing both to the low availability of LC-KA05 and its high instability, it was not possible to carry out co-crystallization experiments). Of the five mutants, only E64Q and R326Q yielded co-complexes with **1**; **7** was not observed in the active sites, as the acetate of **1** was clearly visible. The structure of the LkcE E64Q/LC-KA05 complex was solved at 3.03 Å resolution and that of the LkcE R326Q/LC-KA05 complex at 2.50 Å resolution by molecular replacement, using the structure of the LkcE/EMAA/DATD complex as the search model, and the presence of LC-KA05 confirmed using a weighted $F_o{-}F_c$ omit map (Supplementary Fig. 3; Supplementary Table 3).

LC-KA05 lies at the interface between the substrate-binding domain and the cofactor-binding domain and adopts an extended conformation. The substrate sits in a deep pocket ($\sim 20$ Å from the protein surface and $\sim 6{-}8$ Å wide), and is surrounded by hydrophobic residues (L70, Y168, Y182, L185, M189, W200, L324, F345, V396, and G398). In addition to these hydrophobic interactions, binding is likely mediated by seven hydrogen bonds: two to the lactone ring at the entry point of the pocket (by N179 and T397), one to the acetate at C-7 by the side chain of Q428, and four to the lactoyl moiety (two with Y168, and one each with E64 and R326). Notably, in this configuration, the lactoyl portion of LC-KA05 stacks against the FAD isoalloxazine in a position appropriate for hydride transfer (the amide proton is situated at 3.2 Å from the FAD N-5). On the other hand, superposition of the R326Q/LC-KA05 structure and that of the holo wild type shows that if LC-KA05 bound into the wild type in the same orientation as in the LkcE R326Q/LC-KA05 complex, its C-24 hydroxyl would sterically clash with R326 (Supplementary Fig. 13). In the E64Q structure in which R326 is present, the protein is identical, but the substrate adopts an alternative orientation, which allows it to interact with R326 (3.1 Å between the C-23 carbonyl of LC-KA05 and the R326 side chain) (Supplementary Fig. 13).

In both structures obtained in the presence of **1**, the observed linear configuration of the polyketide chain would not permit subsequent cyclization involving the C-2 and C-18 centers (they are separated by 13.2 Å). This observation is consistent with the idea that LC-KA05 bound into the E64Q and R326Q mutants represents an on-pathway, pre-cyclization conformation. Indeed, the fact that the R326Q mutant retains essentially wild-type activity means that we have not simply captured a non-productive complex with the enzyme. To create the space necessary for LC-KA05 to adopt its cyclization-ready state, the enzyme must undergo a conformational change. Based on our structural analysis, we propose that this involves a hinge movement between the cofactor-binding and substrate-binding domains, made possible by the distinctive dimerization mode of LkcE. The new substrate configuration may be stabilized by a second hydrogen bonding interaction with the bifunctional R326. This new configuration would also position the substrate enol in proximity to E64. Such a large-scale conformational change of the enzyme might account for the observation that soaking **1** with the

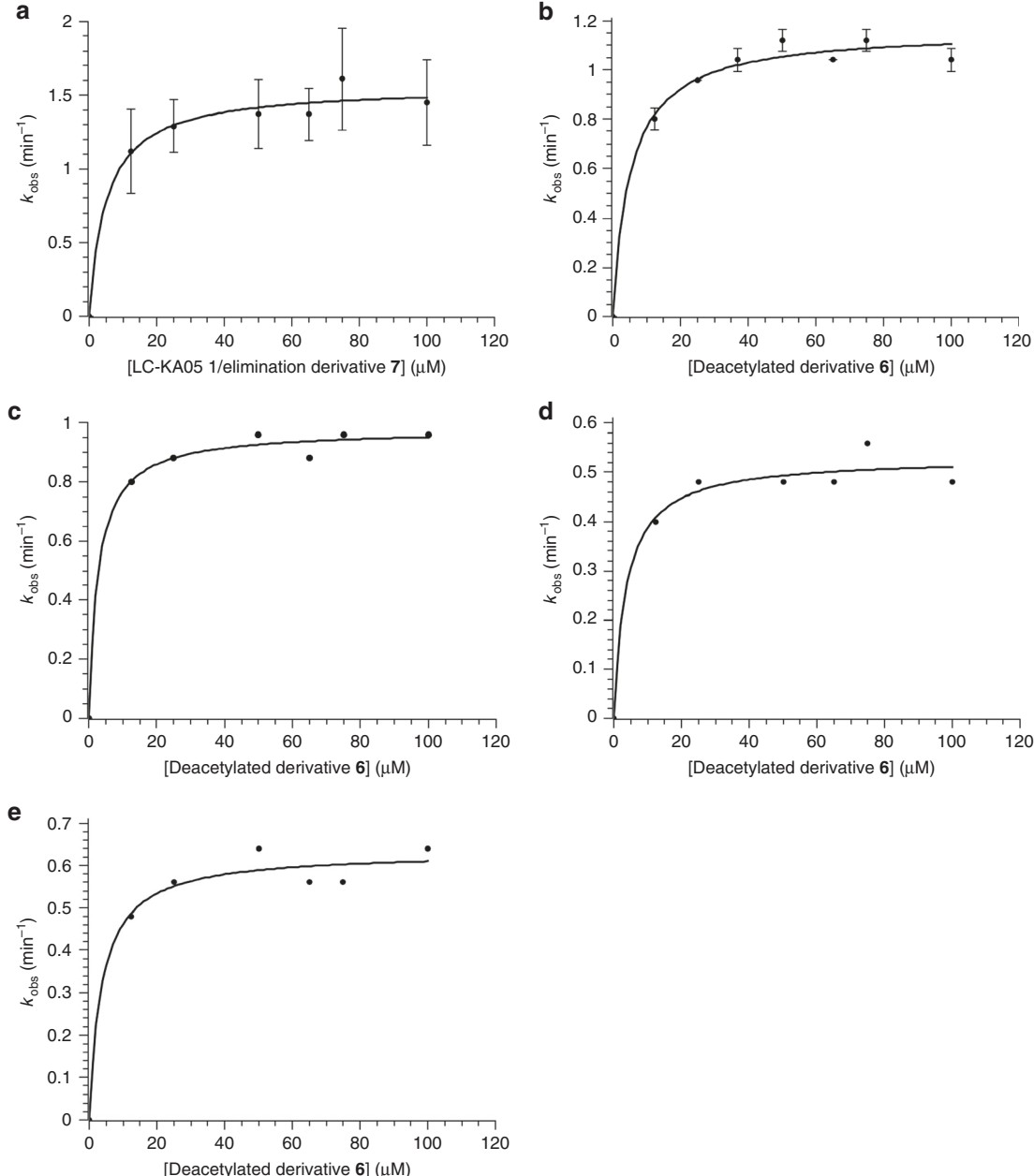

**Fig. 6** Kinetic analysis by coupled assay of LkcE and its mutants. Data for which errors are shown were obtained by experiments repeated in duplicate (**a**) (the error bars thus show the two data points) or triplicate (**b**) (the error bars indicate the standard deviation). **a** LkcE acting on its native substrate, LC-KA05 (**1**), as a 2:1 mixture with eliminated product **7**. **b** LkcE acting on deacetylated (7-OH) substrate **6**. **c** LkcE acting on **6** but with each concentration adjusted to contain the maximum amount of DMSO (that present at 100 μM **6** in **b**). As the data in **b** and **c** are essentially identical, the concentration of DMSO in this range had no effect on the rate. **d** LkcE R326Q acting on **6**. **e** LkcE Y182F acting on **6**. It must be noted that as we were limited in these assays for reasons of sensitivity to higher concentrations of substrate, it is possible that we missed an earlier sigmoidal dependence on concentration, indicative of cooperative behavior between the two LkcE monomers

E64Q and R326Q mutants failed to yield a cyclization-compatible conformation.

Based on these data, we propose the following mechanism for LkcE (Fig. 7). LC-KA05 in its enol form binds into the active site via interactions that include H-bonding between the amide portion and R326. The fact that no degradation products of LC-KA05 are observed in the wild type in vivo strongly suggests that binding into LkcE is closely coupled to release of the intermediate from the PKS and its acetylation, perhaps by physical association of LkcE, the last PKS subunit, LkcG, and the as yet unidentified acetyltransferase. Hydride transfer[9] then occurs to FAD to form

the iminium. Although we initially considered that the conserved R326 in the active site might facilitate substrate oxidation by acting as a catalytic general base to deprotonate the amide[27], the near wild-type activity of the R326Q mutant is instead consistent with the residue playing a role in maintaining the cyclization-ready conformation of the substrate established by a subsequent conformational change. The side chain of E64 would then initiate ring closure by acting as a general base catalyst to activate the enol (there is no counterpart of this residue, to our knowledge, in other MAO family members[9]). This key role is supported by the complete loss of activity seen with the E64A and E64Q mutants.

**Table 1 Kinetic data obtained for LkcE and its mutants**

| Enzyme | Substrate(s) | $k_{cat}$ (s$^{-1}$) | $K_M$ (μM) |
|---|---|---|---|
| LkcE wt | LC-KA05 (1)/elimination derivative (7) mixture | 3.4 ± 0.2 | 5 ± 2 |
| LkcE wt | Deacetylated derivative (6) | 2.4 ± 0.1 | 5 ± 1 |
| LkcE wt | Deacetylated derivative (6) complemented with DMSO | 2.2 | 3 |
| LkcE wt | EMAA (4) | X | X |
| LkcE wt | DATD (5) | X | X |
| LkcE wt | EMAA (4)/DATD (5) | X | X |
| LkcE E64A | LC-KA05 (1)/elimination derivative (7) mixture | X | X |
| LkcE E64Q | LC-KA05 (1)/elimination derivative (7) mixture | X | X |
| LkcE R326L | LC-KA05 (1)/elimination derivative (7) mixture | X | X |
| LkcE R326Q | Deacetylated derivative (6) | 0.88 | 4 |
| LkcE Y182F | Deacetylated derivative (6) | 1.5 | 4 |

X = no detected activity

**Fig. 7** Proposed mechanism of the LkcE-catalyzed reaction. LC-KA05 binds into the active site, and undergoes FAD-catalyzed oxidation to yield the iminium ion. A conformational change then occurs to bring the δ-lactone in proximity to the iminium, with the new LC-KA05 conformation potentially stabilized by interaction with R326. The final step is an intramolecular Mannich reaction, involving general base catalysis by E64 to generate the nucleophilic C-2 enolate

This mechanism could also explain the advantage of modifying the intermediate with an acetyl protecting group[28] as this assures that attack on the iminium occurs only from C-2 and not from the C-7 hydroxyl.

## Discussion

Studying how enzymes naturally gain new functions is a valuable source of information for attempts to replicate this process in the laboratory using rational design or combined rational/directed evolution approaches[29]. Here, we have investigated the bifunctional amide oxidase/macrocyclase LkcE to understand how a subsequent cyclization function might have been acquired by an MAO scaffold.

Existing structure/function data, although insufficient to reliably trace the evolution of the MAO family, clearly demonstrate that the shared di-domain structure of the enzymes can support a common oxidation mechanism of diverse amine substrates, including primary and secondary amines, polyamines, amino acids, and methylated lysine side chains in proteins[9]. Our results strongly suggest that the new catalytic activity of LkcE depends, among other factors, on a change in the mode of dimerization of the enzyme relative to classical members of the MAO family, such as hMAOs A and B. With dimerization limited to the substrate-binding domain in LkcE, the cofactor-binding domain is substantially less constrained. This physical decoupling would appear to permit the two domains to undergo a hinge motion, which we propose underlies a necessary conformational change of the initially-bound substrate between an essentially linear and a cyclization-ready conformation. In addition, the residue at position 326 (LkcE numbering), which in the hMAOs is positioned at the dimer interface, is liberated in LkcE, allowing it to serve a

catalytic role in the active site. E64, a second residue that has been implicated in the reaction mechanism, is located in a position that favors non-deleterious mutation, on a surface loop whose sequence is highly divergent between LkcE and hMAOs A and B (Supplementary Fig. 2).

Evolving specificity for LC-KA05 relative to classical MAO substrates also evidently required a substantial shift in amino-acid sequence within the substrate-binding domain (Supplementary Fig. 2)[1], as well as at least two mutations increasing the space available adjacent to the FAD cofactor. In this context, the inherent promiscuity of the MAOs likely provided a favorable evolutionary starting point for the acquisition of new substrate specificity[1]. We have shown here that LkcE is capable of binding two substantially smaller substrate analogs, and shows tolerance toward structural variation at C-6/C-7 in the native substrate. Although further work will be required to define its substrate profile in detail, these data encourage the idea that LkcE may find wider use as a ligation/macrocyclization catalyst in both synthetic biology and organic synthesis applications. In view of the largely hydrophobic nature of the binding pocket, it will be particularly interesting to explore whether minimally functionalized acyl chains are also substrates.

In conclusion, on the basis of our analysis we propose that the bifunctional amide oxidase/macrocyclase LkcE acting on a highly functionalized polyketide chain, arose from an ancestral amine oxidase not only through substantial modification of the substrate-binding region and recruitment of two catalytic residues into the active site, but also by adoption of a different quaternary organization from modern MAOs. This adds to the emerging awareness of the wider structural alterations that may be necessary to introduce new reaction chemistry into existing enzyme active sites[30].

## Methods

**Analysis in silico of LkcE**. As a starting point for characterizing LkcE, we identified its closest sequence homologs in the NCBI database using BlastP[31]. For the top 11 hits, we also determined the genomic context of the genes (Supplementary Fig. 2). This analysis revealed nine LkcE homologs located within complete or partial lankacidin (or the closely related chejuenolide) gene clusters (as determined with reference to that described in *S. rochei*) (Supplementary Fig. 2), with sequence identity to the *S. rochei lkcE* gene in the range of 54−100%. To aid in defining features that distinguish LkcE from classical monoamine oxidases, we also included in our alignment the two nearest homologs which are not present in lankacidin clusters. We also used two of the closest structural homologs to LkcE from the PDB, which show amine oxidase activity (as identified using the DALI[13] server). Multiple sequence alignment was carried out using ClustalW (https://npsa-prabi.ibcp.fr/cgi-bin/npsa_automat.pl?page=/NPSA/npsa_clustalw.html)[32] and the figures created using ESPript[33].

**Materials and DNA manipulation**. Biochemicals and media were purchased from Thermo Fisher Scientific (EDTA), VWR (glycerol, NaPi, NaCl), BD (peptone, yeast extract), Euromedex (isopropyl β-D-1-thiogalactopyranoside; IPTG), and Sigma-Aldrich (imidazole). The enzymes for genetic manipulation were purchased from Thermo Fisher Scientific and NADH peroxidase from NZYTech. DNA isolation and manipulation were performed using standard methods[34,35]. Isolation of DNA fragments from agarose gel and purification of PCR products were carried out using the NucleoSpin Extract II kit (Macherey Nagel, Hoerdt, France). Standard PCR reactions were performed with Phusion high-fidelity DNA polymerase (Thermo Fisher Scientific); reactions were carried out on a Mastercycler Pro (Eppendorf). Synthetic oligonucleotides were purchased from Sigma-Aldrich, and DNA sequencing was carried out by GATC Biotech (Mulhouse, France). All organic solvents used were HPLC grade and purchased from Sigma-Aldrich.

**Bacterial strains and culture conditions**. *E. coli* strain DH5α was used for cloning (the primer sequences are provided in Supplementary Table 1), *E. coli* Rosetta 2 and *E. coli* B834 pRARE2 (DE3) were used respectively for producing unlabeled and seleniated protein, *E. coli* BW25113 for PCR targeting[22,36], and *E. coli* ET12567 [pUZ8002] for transforming *S. rochei* var. *volubilis* ATCC 21250. This strain of *S. rochei* was a kind gift of Professor P.F. Leadlay (University of Cambridge). *E. coli* strains were grown on 2TY (16 g L$^{-1}$ tryptone, 5 g L$^{-1}$ yeast extract, 5 g L$^{-1}$ NaCl, adjusted to pH 7.6 with NaOH) for cloning purposes and LB (10 g L$^{-1}$ tryptone, 10 g L$^{-1}$ yeast extract, 5 g L$^{-1}$ NaCl, adjusted to pH 7 with NaOH) for production of unlabeled protein. Seleniated protein was produced in M9 medium (50 mM Na$_2$HPO$_4$, 22 mM KH$_2$PO$_4$, 10 mM NaCl, 20 mM NH$_4$Cl, adjusted to pH 7.2 with NaOH). After autoclaving, sterile-filtered ingredients were added as follows: 50 mg L$^{-1}$ of thiamine and riboflavin, 4 g L$^{-1}$ glucose, 100 μM CaCl$_2$, 2 mM MgSO$_4$, 40 mg L$^{-1}$ selenomethionine, and 40 mg L$^{-1}$ of the remaining 19 amino acids). The *E. coli* cultures also contained the appropriate concentration of antibiotics (50 mg mL$^{-1}$ kanamycin and 25 mg mL$^{-1}$ chloramphenicol in pre-cultures; 5 and 2.5 mg mL$^{-1}$, respectively, in protein production cultures). *S. rochei* was grown on SFM (20 g L$^{-1}$ mannitol, 20 g L$^{-1}$ soy flour, 20 g L$^{-1}$ agar) in order to produce a spore suspension for conjugation with *E. coli*, and yeast extract-malt extract-glucose (YMG) medium (4 g L$^{-1}$ glucose, 4 g L$^{-1}$ yeast extract, 10 g L$^{-1}$ malt extract, adjusted to pH 7.2 with KOH) for metabolite production. For conjugation experiments, a spore suspension was mixed with the *E. coli* 12567[pUZ8002] strain and plated on SFM containing 10 mM MgCl$_2$. For metabolite production, *S. rochei* was grown at 30 °C with shaking at 180 rpm in a rotary incubator for 3 days.

**Sub-cloning and site-directed mutagenesis of *lkcE***. Plasmid pGEX-LcsJ (a kind gift of Professor P.F. Leadlay; *lcsJ* is equivalent to *lkcE*[23]) was digested with *Bam*HI and *Xho*I, and the resulting fragment cloned into the equivalent sites of plasmid pBG-102 (Center for Structural Biology, Vanderbilt University), which codes for a His$_6$-SUMO tag, to yield pBG-102-LkcE (the full list of plasmids used in this study is provided in Supplementary Table 2). The sequence of *lkcE* was re-verified by sequencing. Site-directed mutations were introduced into *lkcE* by PCR using mutagenic oligonucleotides (Supplementary Table 1) and Phusion high-fidelity polymerase, followed by digestion of the parental DNA by 1 μL of *Dpn*I Fast digest (Thermo Fischer Scientific). The presence of the correct mutations was confirmed by sequencing.

**Expression and purification of recombinant LkcE and mutants**. After an overnight pre-culture at 37 °C, *E. coli* Rosetta 2 containing pBG-102-LkcE was grown in LB medium supplemented with riboflavin (10–50 mg L$^{-1}$) in order to support FAD biosynthesis. When the cultures reached an A$_{600}$ of 0.6, the cultures were subjected to a combined chemical and thermal shock (addition of 3% ethanol, followed by 2 h at 4 °C), and then protein expression induced by the addition of IPTG (final concentration of 0.1 mM). Incubation was then continued at 15 °C overnight. The culture was then centrifuged for 30 min at 3500 *g*, the resulting cell pellet resuspended in 30 mL phosphate buffer (100 mM NaPi, 10% glycerol, 10 mM EDTA, pH 7.4), and the cells re-centrifuged, before storage at − 20 °C.

The cells resulting from 1 L of culture were resuspended in lysis buffer (30 mM 4-(2-hydroxyethyl)-1-piperazineethanesulfonic acid (HEPES), 500 mM NaCl, pH 7.5) (10 mL of buffer were used for each A$_{600}$ unit at the end of culturing). In total, 400 units of benzonase were added to each 100 mL of culture, as well as 6 mM MgSO$_4$, in order to eliminate nucleic acids. The cells were then lysed using a cell disruptor (Basic Z, Constant Systems Ltd.) at 15 kPsi (1000 bars) at 4 °C, and the cellular debris removed by centrifugation (35,000 *g* for 40 min). After sterile filtration (0.22 μm filter) and addition of 70 mM imidazole, the supernatant was injected at 3 mL min$^{-1}$ onto a Ni-Sepharose column (GE Healthcare Life Sciences) pre-equilibrated in lysis buffer containing 70 mM imidazole, using an Äkta Avant system (GE Healthcare Life Sciences). This was followed by a wash step using equilibration buffer until the OD stabilized. The protein was then eluted using the same buffer but containing 250 mM imidazole, at 5 mL min$^{-1}$. The LkcE-containing fractions were identified by 12.5% sodium dodecyl sulfate–polyacrylamide gel electrophoresis (SDS-PAGE), and pooled. The His$_6$-SUMO tag was removed by incubation with 150 μg human rhinovirus 3 C protease (HRV3C protease), and dialyzed against lysis buffer overnight to eliminate the imidazole. The sample was then reinjected onto the Ni-Sepharose column (2 mL min$^{-1}$), which had been pre-equilibrated with lysis buffer containing 20 mM imidazole, and the flow-through containing LkcE collected. After this reverse nickel step, the protein was diluted three times into reduced ionic strength buffer (30 mM HEPES, 1 mM EDTA, pH 7.5), in order to allow for purification by anion exchange.

For this, the sample was injected (5 mL min$^{-1}$) onto a Q-sepharose column (trimethylammonium on 6% agarose) equilibrated in buffer (30 mM HEPES, 100 mM NaCl, 1 mM EDTA, pH 7.5). LkcE was then eluted using a NaCl gradient (from 100 mM to 1 M) at 5 mL min$^{-1}$. The LkcE-containing fractions were identified by 12.5% SDS-PAGE, and pooled. In order to eliminate the remaining contaminants and any aggregates, the protein was subjected to a final gel filtration step. For this, it was concentrated using an Amicon Ultracel-30 (Merck Millipore) by centrifugation at 4000 *g*, to obtain a volume of less than 8 mL. This was then injected via a 10 mL loop onto a Superdex 200 26/60 prep grade column (GE), which had been pre-equilibrated in buffer (30 mM HEPES, 150 mM NaCl, 1 mM EDTA, pH 7.5). The protein was eluted in the same buffer at 2 mL min$^{-1}$. Fractions containing pure LkcE were identified by 12.5% SDS-PAGE, and pooled. The protein was then concentrated to 20 mg mL$^{-1}$ by centrifugation at 4000 *g* using an Amicon Ultracel-30, and 50 μL aliquots frozen in liquid nitrogen prior to storage at −80 °C.

**Analysis by CD of LkcE and its mutants**. CD was carried out on a Chirascan CD (Applied Photophysics). Data were collected at 0.5 nm intervals in the wavelength range of 200−260 nm at 20 °C, using a temperature-controlled chamber. A 0.01 cm cuvette containing 30 μL of protein sample at 50 μM was used for all the measurements. Each spectrum represents the average of three scans, and sample spectra were corrected for buffer background by subtracting the average spectrum of buffer alone.

**Identification of LkcE FAD cofactor**. The LkcE cofactor was identified as FAD by passing the enzyme over a reverse phase C8 column (Grace) using an HPLC (Äkta Explorer, GE Healthcare) in a gradient of 0−80% acetonitrile containing 0.1% trifluoroacetic acid. The peaks corresponding to the protein and to the released cofactor were then analyzed by mass spectrometry (F. Dupire, Mass Spectrometry Service of the Faculty of Sciences and Technologies, Université de Lorraine). The FAD content of LkcE was then estimated by UV-Vis by measuring its absorbance (450 nm, $\varepsilon = 11500$) using a SAFAS spectrometer (UVmc$^2$), as well as that of the protein (280 nm, $\varepsilon = 56840$). For this, LkcE was prepared at three different concentrations, and the OD measured at 280 and 540 nm. The value at 280 nm was divided by 56,840 to obtain the protein concentration and that at 540 nm by 11,500 to obtain the FAD concentration, and then the ratio of the two values determined. The % FAD values reported represent the average of the three calculated concentrations.

**Creation of the *lckE* knockout strain of *S. rochei***. The gene *lkcE* was inactivated in *S. rochei* var *volubilis* ATCC 21250 by PCR targeting, using a method similar to that described previously[8]. For this, oligonucleotides (Supplementary Table 2) were designed to amplify an apramycin resistance cassette, flanked on both sides with 39 bp of homology to the genomic regions up and downstream of *lkcE*. These primers were used in a PCR reaction with plasmid pIJ773 (ref. [22]) encoding for Apra$^R$. The resulting PCR product was then used to replace the *lkcE* region in cosmid Lc2B12 (kind gift of Professor P.F. Leadlay), a derivative of SuperCos1 (ref. [23]). For this, *E. coli* BW25113 was co-transformed with cosmid Lc2B12, plasmid pIJ790 (temperature-sensitive λRed recombination helper plasmid[22]), and the PCR fragment. Recombinants were selected by growth on LB supplemented with apramycin, and the presence and correct location of the deletion confirmed by PCR screening and sequencing. *E. coli* ET12567[pUZ8002] was then transformed with the mutant cosmid, and used for conjugation with spores of *S. rochei*. Recombinants were selected for apramycin resistance, and the inactivation of *lkcE* on plasmid pSRV (this plasmid contains the lankacidin cluster[8]) was confirmed by sequencing. The presence of the intermediate (and the absence of later-stage metabolites) in the *lkcE* mutant was confirmed by HPLC-MS analysis, by comparison with the wild-type strain (Supplementary Fig. 8 and Supplementary Table 4).

**Isolation of LC-KA05**. The *lkcE* mutant of *S. rochei* was grown in YMG medium (3 × 10 L of culture). After removal of the cells by centrifugation (4550 *g*, 30 min), the supernatant was extracted with ethyl acetate (3 × equivalent volume). The organic phase was then evaporated, yielding 100−200 mg of material. Approx. 200 mg crude extract was dissolved in 2 mL methanol and cleared by centrifugation (2 min, 9612 *g*). The clear supernatant was purified by preparative HPLC on a Nucleodur C$_{18}$ Isis column (5 μm, 250 × 21 mm; Macherey Nagel) using a linear gradient (0 min: 80% A, 20% B; 100 min: 55% A, 45% B; flow = 15 mL min$^{-1}$) of water (A) and acetonitrile (B). The chromatography was monitored by electrospray ionization (ESI)-MS using a 1:500 static splitter (methanol was used to achieve splitting at a flow rate of 500 μL min$^{-1}$) coupled to a ZQ mass spectrometer (capillary voltage 3 kV, 650 L h$^{-1}$ nitrogen, 250 °C desolvation temperature; Waters). LC-KA05 showed a typical retention time of 39−48 min. The fractions were pooled and freeze dried yielding 20 mg as a 2:1 mixture with an elimination product (C$_{25}$H$_{35}$NO$_6$) **7**, corresponding to loss of the C-7 acetate group. It should be noted that traces of formic acid in the chromatography solvents lead to quantitative formation of the elimination product upon freeze drying. An additional hydrolysis product **6** was detected (free C-7-OH group) at a retention time of 23−27 min. Assignment of the identities of **6** and **7** was based on HR-ESI-MS data (QToF Premier (Waters)), as well as their fragmentation patterns (ESI positive), which show characteristic peaks in common with each other and with **1** (Supplementary Table 4). The $^1$H-NMR spectrum of **6** showed the absence of H-27, while the difference NMR spectrum of **7** also revealed the absence of signals corresponding to H-6, H-6′ and the C-7 acetate (H-27), as expected (Supplementary Fig. 9).

Spectral data for LC-KA05 (**1**) (and see Supplementary Fig. 9): $^1$H-NMR (400 MHz, CD$_3$OD) δ: 6.35 (d, *J* = 15.4 Hz, 1H, *H*-9), 6.25 (d, *J* = 15.6 Hz, 1H, *H*-15), 5.71 (dd, *J* = 6.6 and 15.6 Hz, 1H, *H*-14), 5.64−5.45 (m, 4H, *H*-7, *H*-8, *H*-11 and *H*-17), 4.21−4.16 (m, 2H, *H*-5 and *H*-13), 4.12 (q, *J* = 7.0 Hz, 1H, *H*-24), 4.02−3.92 (m, 2H, *H*-18 and *H*-18′), 2.50−2.36 (m, 3H, *H*-4, *H*-12 and *H*-12′), 2.22 (ddd, *J* = 6.2, 8.9 and 14.5 Hz, 1H, *H*-6), 2.05 (s, 3H, *H*-27), 1.91 (ddd, *J* = 4.4, 6.6 and 14.5 Hz, 1H, *H*-6′), 1.83 (s, 3H, *H*-22), 1.76 (s, 3H, *H*-21), 1.73 (s, 3H, *H*-16), 1.35 (d, *J* = 7.0 Hz, 3H, H-25), 1.27 (d, *J* = 7.2 Hz, 3H, *H*-20) ppm. $^{13}$C-NMR (100 MHz, CD$_3$OD) δ: 176.2 (C-23), 170.7 (C-26), 169.6 (C-3), 168.8 (C-1), 138.3 (C-9), 135.5 (C-16), 134.3 (C-10), 133.9 (C-15), 130.8 (C-14), 129.7 (C-11), 127.2 (C-17), 123.6 (C-8), 96.7 (C-2), 77.7 (C-5), 72.4 (C-7), 71.8 (C-13), 67.7 (C-24), 37.9 (C-18), 36.8 (C-4), 36.5 (C-6), 36.2 (C-12), 19.9* (C-25/C-27), 19.8* (C-25/C-27), 15.4 (C-20), 11.3 (C-22, C-21), 7.2 (C-19). *: These signals cannot conclusively be assigned. HR-ESI-MS: 528.2575 (calc. for C$_{27}$H$_{39}$NO$_8$Na$^+$: 528.2573).

Spectral data for the elimination product **7**: HR-ESI-MS: 468.2356 (calc. for C$_{25}$H$_{35}$NO$_6$Na$^+$: 468.2362) and see Supplementary Fig. 9 and Supplementary Table 4.

Spectral data for the purified C-7 hydrolysis product **6**: HR-ESI-MS: 486.2462 (calc. for C$_{25}$H$_{37}$NO$_7$Na$^+$: 486.2468) and see Supplementary Fig. 9 and Supplementary Table 4.

**Crystallization and X-ray data collection**. For protein crystallization, LkcE was purified and stored in buffer (30 mM HEPES, 100 mM NaCl, 1 mM EDTA, pH 7.5) at a final concentration of 20 mg mL$^{-1}$. Homogeneity was checked by dynamic light scattering using a Zetasizer NanoS (Malverne). Native LkcE crystals were produced with the JCSG + kit (Molecular Dimensions), under conditions of 15% PEG 8000, 160 mM calcium acetate, 20% glycerol, 80 mM sodium cacodylate, pH 6.5. The 3 μL drops contained a 2:1 mixture of protein solution (5 mg mL$^{-1}$, 10 mM DATD, 10 mM EMAA, and 1 mM TCEP). Crystals of SeLkcE were obtained using the hanging drop method in Linbro® plates under conditions of 28% PEG 3350, 200 mM ammonium acetate, 100 mM Bis Tris, pH 6.5. This condition was identified using the INDEX screen (Hampton Research). Drops were formed by mixing 2 μL of a protein solution (8 mg mL$^{-1}$ SeLkcE, DATD 5 mM) with 1 μL of crystallization buffer. Crystals grew in 2−3 days. Crystals of mutants E64Q and R326Q were obtained by microseeding using a Seed Bead kit (Hampton Research). Drops contained 1 μL of crystal shred in 16% PEG 8000, 160 mM calcium acetate, 20% glycerol, 80 mM sodium cacodylate, pH 6.6, and 2 μL of protein solution (6 mg mL$^{-1}$). Crystals were soaked in crystallization buffer with 30% ethylene glycol before being frozen in liquid nitrogen. The crystals of mutants were additionally soaked in cryoprotection buffer containing 30 mM LC-KA05, before cryo-cooling. X-ray diffraction data on SeLkcE and native/mutant LkcE were collected at the SOLEIL synchrotron on the Proxima2 and Proxima1 beamlines, respectively. The images were integrated using X-ray diffraction spectroscopy in the space group $P4_12_12$.

**Structure determination and refinement**. As LkcE shares only 25% sequence identity with the closest homolog in the PDB (6HDNO (PDB 3NG7)[14]) we acquired a complete multiple wavelength anomalous diffraction data set on SeLkcE. Initial phases were ultimately generated using SAD using the peak wavelength (λ = 0.9793 nm). Sixteen selenium positions (out of a possible 18) were located with the SHELX[37] package. An initial model was built automatically using BUCANEER[38,39] with several cycles of manual rebuilding in Coot[40] and refinement with Buster[41]. The structures of native LkcE and mutants were solved by molecular replacement with Phaser MR[42] using the structure of SeLkcE as the search model, manually

rebuilt in Coot and refined with Buster. All the structures were validated using the program MolProbity[43]. All protein structure figures were prepared using the program PyMol (Schrödinger, LLC). Data collection, refinement, and validation statistics are presented in Supplementary Table 3.

**LkcE activity tests in vitro with substrate analogs**. LkcE (50 μM) was incubated with 1, 5, or 50 mM of the two substrate analogs (in combination or separately) at 25 °C for 1 h in buffer (30 mM HEPES, 150 mM NaCl, 1 mM EDTA, pH 7.5), and the reaction was stopped by adding 2 × the equivalent volume of ethyl acetate. The organic phase was evaporated overnight at room temperature and then the residue analyzed by HPLC-MS (see section 2.1.3), after resuspension in H$_2$O/acetonitrile (80:20 v/v).

**LkcE activity tests in vitro with the native substrate**. Steady-state kinetics parameters were determined at 25 °C in gel filtration buffer (30 mM HEPES, 150 mM NaCl, 1 mM EDTA, pH 7.5) with variable concentrations of substrate, either the LC-KA05 (**1**)/elimination derivative product **7** mixture, the deacetylated derivative **6**, or the substrate analogs EMAA (**4**) and DATD (**5**) (in combination and separately). Reactions were carried out with 2 μM LkcE (and additionally with 4 μM enzyme in experiments with the substrate analogs **4** and **5**) in the presence of NADH peroxidase from *S. faecalis* (0.5 U mL$^{-1}$) and NADH (0.3 mM). Initial rate measurements were obtained on a SAFAS UVmc$^2$ spectrophotometer by following the oxidation of NADH at 340 nm. Where appropriate, data were fitted to the Michaelis–Menten equation using least-squares regression analysis to determine $k_{cat}$ and $K_M$. We confirmed that the NADH peroxidase was not rate-limiting in the reaction by showing that doubling its concentration (from 3 to 6 U mL$^{-1}$) had no effect on the rate, whereas increasing the concentration of LkcE by a factor of five (from 5 to 250 μM) produced the expected fivefold increase in velocity.

Large-scale assays for analysis by HPLC-MS were carried out with 20 μM active LkcE or denatured enzyme (obtained via incubation for 10 min at 95 °C) in buffer (30 mM HEPES, 150 mM NaCl, 1 mM EDTA, pH 7.5). In total, 100 μM substrate (LC-KA05 **1**/elimination derivative **7** mixture or deacetylated derivative **6**) was added every 10 min, until a final concentration of 1 mM substrate was reached. Enzymatic reactions were carried out overnight at 25 °C. One milliliter of ethyl acetate was then added and the mixture was thoroughly vortexed. The organic phase was separated from the aqueous phase, and the extraction repeated twice. The combined organic layers were evaporated to dryness on a SpeedVac concentrator.

**HPLC-MS analysis**. HPLC-MS analysis was performed using an HPLC (Dionex, Ultimate 3000) coupled to a LTQ Orbitrap XL hybrid mass spectrometer (Thermo Scientific) fitted with an ESI source. HPLC-MS data were processed using Xcalibur (v. 2.1) (Thermo Scientific). For analysis of the compounds of interest, the HPLC was fitted with an Alltima 5μ C18 column (150 mm × 2.1 mm, Grace Alltech) column. A solvent system of acetonitrile and water both containing 0.1% formic acid (v/v) was used. Samples were first eluted with a linear gradient from 20−35% acetonitrile over 20 min, followed by a second linear gradient from 35−90% acetonitrile over 15 min, at a flow rate 0.2 mL min$^{-1}$. The mass spectrometer was run in either positive or negative ionization modes scanning from *m/z* 100−1000. Mass spectrometric conditions were as follows for ESI$^+$ mode: spray voltage was set at + 4.5 kV; source gases were set (in arbitrary units min$^{-1}$) for sheath gas, auxiliary gas, and sweep gas at 30, 10, and 10 respectively; capillary temperature was set at 275 °C; capillary voltage at + 4 V; and tube lens, split lens, and front lens voltages at +155 V, −28 V, and −6 V, respectively. The MS parameters for the ESI$^-$ mode were directly transposed from the ESI$^+$ mode. To confirm some structures, MS$^2$ fragmentations in the ion trap and exact mass scans using the Orbitrap analyzer were carried out.

## Data availability

Protein Data Bank coordinates for seleniated *holo* LkcE wild type, LkcE wild type in complex with EMAA (**4**) and DATD (**5**), and the LkcE E64Q and R326Q mutants in complex with LC-KA05 (**1**), have been deposited under accession codes 6FJH, 6F32, 6F7V, and 6F7L, respectively. All data presented in the manuscript are available upon reasonable request from the corresponding authors.

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

## Acknowledgements

Funding for this work was provided by the Agence Nationale de la Recherche (ANR-11-JSV8-003-01 and ANR-16-CE92-0006-01 to K.J.W.), the Centre National de la Recherche Scientifique (CNRS), the 'IMPACT Biomolecules' project of the Lorraine Université d'Excellence (Investissements d'avenir—ANR 15-004), the Lorraine Region (Bonus Qualité Recherche (BQR) grants to K.J.W. and B.C.) and the Deutsche Forschungsgemeinschaft (Cluster of Excellence REBIRTH, "From Regenerative Biology to Reconstructive Therapy" EXC 62 (to G.D. and A.K.), and grant Ki 397/20-1 (to A.K.)). Jean-Christophe Lec is thanked for invaluable assistance with the kinetics analysis, Russell Cox for helpful discussions, and Peter Leadlay for suggesting the use of EMAA and DATD and for editing of the manuscript. We also acknowledge the following for help in data acquisition: Pierre Legrand and Andrew Thompson (Soleil Synchrotron, Beamline Proxima1), William Shepard and Martin Savko (Soleil Synchrotron, Proxima2), and Javier Perez and Aurelien Thureau (Soleil Synchrotron, Swing). A portion of the NMR data was recorded on the NMR spectrometer of the Plateforme de Biophysique et Biologie Structurale (B2S) (IBSLor, UMS2008, CNRS-UL-INSERM).

## Author contributions

J.D. cloned, expressed and purified LkcE, crystallized LkcE, solved the structures, and analyzed data (along with A.G.). F.R. carried out the kinetic analysis of wild-type LkcE and mutants. C.J. generated the LkcE knockout and cultured and extracted the resulting mutant. S.C. contributed to isolation of LC-KA05 (along with B.C., J.D., and A.G.) and the crystallography. G.D. and A.K. purified and structurally characterized LC-KA05. C.P. carried out analysis of the lankacidin metabolites by HPLC-MS, and helped with data interpretation by K.J.W. A.G. and K.J.W. designed the study,

supervised the research, and analyzed the data. K.J.W. wrote the paper, with input from all other authors.

## Additional information

**Competing interests:** The authors declare no competing interests.

