## [Peer Review File · Nature Communications]

Reviewers' comments:

Reviewer #1 (Remarks to the Author):

In this manuscript, Dorival et al (Weissman lab) reported the biochemical characterization of LkcE, a unique dual-functional monomeric oxidase (MAO) involved in lankacidin biosynthesis. Using X-ray crystallography, structure-guided mutations, and enzyme assays, the authors identified residues important for the dual amide-oxidation and Mannich catalytic activities, as well as a conformational change and dimerization that are key to the catalytic activities. The above results suggest that both structural and catalytic components are needed for new enzyme design, and can likely be applied to synthetic biology. Because of the unique dual function of LkcE amongst the MAO family enzymes, as well as the potential application to a new cyclization strategy for synthetic biology, the high quality of work and its scientific merit is suitable for publication in Nature Communication. However, the main and SI figures require extensive revision as suggested below:

1. Page 2, first paragraph of introduction. More discussion on the introduction of MAO family enzymes and the evolution of MAO family should be included here.
2. Fig. 1. (1) The coloring will confuse the readers about which enzymes consist of which modules (especially because LksC is used differently in different iteration), as well as the reaction sequences. Suggest adapting the format similar to Fig 2 reported by Tatsuno et al in the original J. Antibiotics paper, and add the module number underneath the domain cartoon. (2) The conversion from 1 to 2 is key to this paper. Add the carbon numbers for cyclization and color the carbon numbers in red.
3. Page 2. Lkc gene cluster. Add more discussion here about LkcE, and delete other non-LkcE enzymes. For example, just say that post-PKS, compound 1 is produced, no need to go into details about TE or acetylation. More stress should be put on the novelty of LkcE.
4. Page 3, first paragraph. Cite Fig. 1 and 5 at the end of first sentence.
5. Page 3, structure paragraph. Report the occupancy of FAD here and compare FAD-binding residues with other MAOs.
6. Discussion. Need to add one paragraph discussing that given the highly hydrophobic nature of the substrate binding pocket, how LkcE can be engineered for synthetic biology.
7. Fig. 2. Add in SI the mechanism about how compounds 4 to 9 are formed.
8. Table S3. (1) The R_{free} for wild type is higher than 0.25. Need to explain in SI or add a footnote under Table S3. (2) The B factors of protein and ligand are unusually high. Need to discuss in main text (page 3). (3) Add Ramachandran parameters.
9. All protein figures. Keep the orientation the same when possible to help the readers acclimatize to the structure.
10. Fig. 3. (1) Each monomer should be shown in a different color, for example, deep blue vs light blue, deep purple vs light purple. (2) Distances should be shown along the dotted line. (3) Fig 3D, use Pymol to set the slice thinner so that the depth contrast is more obvious.
11. Fig. 4. Change the view to side by side. Panel A for LkcE dimer, then panel B for hMAO B and panel C for 6HDNO. Then explain in the figure legends that these three were aligned in monomer A.
12. In the SI: add one figure with two panels: (1) SA-Omit map of protein active site residues, (2) SA-Omit map of FAD, and (3) SA-Omit map of ligands.
13. Fig. S2. Make the superposition stereo.
14. Fig. S4. Active site residues and dimer interface residues should be marked.
15. Fig. S8. Draw chemical structures (and number them) on the HPLC diagrams to help the readers understand which peaks correspond to which products.
16. Fig. S9. There should be Chemdraw structures of compounds 1 and 7 with 2D NMR correlation marked in arrows.
17. Fig. S10-12. Label the compound numbers in color on the HPLC peaks.
18. Fig. S13. Suggest changing to double reciprocal. As is, it looks bad without the pre-saturation data points. Also, it should be discussed that without the pre-K_m points, the kinetic curve may miss the sigmoidal curve, an indication of allosteric interactions between two monomers.

Reviewer #2 (Remarks to the Author):

The manuscript by Weissman and colleagues describes the structural and functional analysis of LckE, a monoamine oxidase enzyme involved in the maturation of the hybrid PKS/NRPS natural product lankacidin. The enzyme catalyzes the oxidation of an amide, followed by the subsequent carbon-carbon bond forming step characterized as an intramolecular Mannich reaction. The structure of the native protein is determined by single wavelength anomalous methods, characterized, and compared to other MAO family members. The substrate for LckE is isolated from a mutant strain, characterized, and used in both kinetic and structural studies, supporting a structural mechanism that involves a significant conformational change in the enzyme to enable the intramolecular cyclization step to occur. This is an interesting enzyme that exists within an equally interesting biosynthetic pathway.

There are a few minor issues that preclude complete evaluation at this stage. I also have some suggestions to make some of the data more accessible.

1. The structural data are sound. The structures, while moderate resolution, show good statistics that suggest the structural conclusions are warranted. The authors provide preliminary validation reports that also do not raise any concerns. However, the preliminary validation reports are not completed for the LckE substrate ligand, molecule DRG. More important, no electron density is provided for any of the ligands in Figure 3b-d. For a large, flexible ligand such as used here, it is common for the density to be less than ideal. Particularly at the resolution of the current structures, the authors should provide experimental evidence for the structural conclusions concerning the ligands in the main text.

2. Assuming it does not conflict with figure/table limits of the journal, the mutagenesis data should be moved to the main text. These are some of the more significant results of the paper and they should not be relegated to the SI. (The current figure 4, showing the oligomerization and interface, is less important than the functional data and could be moved to the SI). Regardless of whether they are kept in the SI or moved to the text, the five initial velocity plots should be presented on a single page, allowing better comparison. Finally, I would prefer to see the kinetic data also presented in a table in addition to just the text.

3. The authors should also consider whether it might be more effective to divide the LC-MS data of the SI into separate figures. This would allow the relevant legend to be presented on each page with the panels that describe each experiment. Figure S8 could be divided with panels a-e, f-h, i-m, n-r, and s-w as currently organized but with the legend present on the same page as each. The same could be done for Figures S10 and S12. In figure S12, each set of UV/LCMS-XID/and MS spectra should also contain a panel (perhaps above each series of panels) showing the chemical reaction for what is being performed. For example, on page s35 with panels a-e, the top of the page should show: the substrates, the inactivated enzyme. The page s36 with panels f-j should show the same substrates, the active enzyme, and the product Lankacidinol A (2). As with kinetics above, this is critical for the manuscript and the authors may wish to consider whether an extracted summary of these data can form a figure within the main body of the manuscript.

4. The authors propose an interesting evolutionary pathway toward the new activity involving a large conformational change. While possible, I am not sure that this is sufficiently supported to allow statements like "Our results show that the new catalytic activity depends intimately on a change in the mode of dimerization of the enzyme relative to classical MAO family..." (pg 10). This may be true; however, there may be many features that are required for this new activity.

Responses to the referees' comments:

We appreciate the constructive criticisms of the two referees. Please find our detailed responses below. All changes to the main text and SI have been introduced in red.

We would also note at this stage that we have reintroduced two sentences (page 9) that had been inadvertently omitted from the original version of the ms (as communicated previously), concerning the pertinence of the observed linear conformation of LC-KA05 in the LkcE mutants.

Reviewer #1 (Remarks to the Author):

In this manuscript, Dorival et al (Weissman lab) reported the biochemical characterization of LkcE, a unique dual-functional monomeric oxidase (MAO) involved in lankacidin biosynthesis. Using X-ray crystallography, structure-guided mutations, and enzyme assays, the authors identified residues important for the dual amide-oxidation and Mannich catalytic activities, as well as a conformational change and dimerization that are key to the catalytic activities. The above results suggest that both structural and catalytic components are needed for new enzyme design, and can likely be applied to synthetic biology. Because of the unique dual function of LkcE amongst the MAO family enzymes, as well as the potential application to a new cyclization strategy for synthetic biology, the high quality of work and its scientific merit is suitable for publication in Nature Communication. However, the **main and SI figures require extensive revision as suggested below:**

1. Page 2, first paragraph of introduction. More discussion on the introduction of MAO family enzymes and the evolution of MAO family should be included here.

Response:

We could not find in the literature a systematic recent study of the evolution of the MAO family, in terms of sequence and/or structure (analogous, for example, to a recent investigation of flavin-dependent mono-oxygenases by the Thornton lab (Mascoti, *et al.* (2016) *J. Mol. Biol.* **428**, 3131)). Indeed, to date, detailed structure/function studies have been published for only a few family members (hMAOs A and B, the L-amino acid oxidases (LAAO), the spermine and polyamine oxidases and the histone lysine demethylase LSD1 (Gaweska & Fitzpatrick (2011) *Biomol. Concepts* **2**, 365)). Thus, such a systematic evolutionary study is not yet feasible.

This point, as well as a more general description of the MAO family, have been added to the discussion (page 11), as we felt this a more appropriate point in the ms than the introduction (as here we have not yet demonstrated that LkcE is a family member).

2. Fig. 1. (1) The coloring will confuse the readers about which enzymes consist of which modules (especially because LkcC is used differently in different iteration), as well as the reaction sequences. Suggest adapting the format similar to Fig 2 reported by Tatsuno et al in the original *J. Antibiotics* paper, and add the module number underneath the domain cartoon. (2) The conversion from 1 to 2 is key to this paper. Add the carbon numbers for cyclization and color the carbon numbers in red.

Response:

The reviewer is correct that the original coloring scheme would have been confusing to the non-expert reader. We have thus used consistent coloring to indicate each of the subunits and the discrete enzyme LkcB. On the other hand, we prefer not to adapt the figure to match that of Fig. 2 in Tatsuno, *et al.*, as the mechanism presented in this paper differs from ours, notably in the fact that the authors propose iterative use of the subunit LkcC, necessitating a specific 'programming' of its constituent domains so that they act in select chain extension cycles. We prefer the alternative mechanism illustrated in the figure (Dickshat, *et al.* (2011) *ChemBioChem* **12**, 2408), which accords with the phylogenetic analysis of the KS domains (Nguyen, *et al.* (2008) *Nat. Biotechnol.* **26**, 225), and thus with their substrate specificities.

We have also revised the structures in the figure to match the *Nat. Commun.* ChemDraw template (as well as those in Figs. 2 and 5).

3. Page 2. Lkc gene cluster. Add more discussion here about LkcE, and delete other non-LkcE enzymes. For example, just say that post-PKS, compound 1 is produced, no need to go into details about TE or acetylation. More stress should be put on the novelty of LkcE.

Response:

We would strongly prefer to maintain the discussion about both the TE and the acetylation. We feel it is important to mention the TE explicitly because in typical PKS pathways it is this domain that is responsible for macrocyclization. In lankacidin biosynthesis, this reaction is carried out by LkcE, while the TE catalyses chain release by an alternative mechanism (formation of a δ -lactone). The relatively rare role of LkcE can thus only be appreciated by comparison to more classical pathways. We would also argue that it's important to mention the acetylation, as the native substrate of LkcE incorporates this functional group, but the enzyme was almost equally active with the deacetylated analogue. We have, however, reorganized the introduction so that the full pathway description is presented in the same paragraph (page 3).

In response to the second comment, we have revised the initial description of LkcE as follows (page 3): The **catalytic** mechanism proposed for LkcE involves initial oxidation of the amide function of LC-KA05 to an iminium ion⁸. This is followed by attack of a C-2 enolate anion on the iminium to give the 17-membered lankacidinol A (**2**)⁸ – an enzymatic intramolecular Mannich reaction. **Although numerous FAD-dependent amine oxidases have been characterized to date⁹, to our knowledge LkcE is only the second described amide oxidase¹⁰. This observation, coupled with the fact that no other known enzyme carries out Mannich chemistry except as a promiscuous activity¹¹, motivated our interest in establishing a detailed structure/function relationship for LkcE.**

4. Page 3, first paragraph. Cite Fig. 1 and 5 at the end of first sentence.

Response:

We have cited Fig. 1 as requested, but as Fig. 5 presents information based on the rest of the paper, it does not seem reasonable to reference it at this early stage.

5. Page 3, structure paragraph. Report the occupancy of FAD here and compare FAD-binding residues with other MAOs.

Response:

The FAD occupancy in the structure is 100% (this figure has been added to the paper, bottom page 4/top page 5). Although the purified protein has a lower FAD content, evidently, only those molecules incorporating the co-factor crystallized.

We have also compared the FAD-binding residues in LkcE to its 5 closest homologs cited in the paper. This analysis in fact revealed another point of divergence between LkcE and 4 out of 5 of these enzymes. The following paragraph summarizing our observations has been added to page 5 (which also necessitated that we relocate the paragraph describing the homologs to page 4), and the specific residues are now indicated in Supplementary Fig. 2 (old Supplementary Fig. 4).

Although the specific FAD-binding residues differ among LkcE and its closest structural homologs (Supplementary Fig. 4), the types of interactions are similar, with the exception of those to the isoalloxazine ring. In all of the structures except LkcE and PPOX, the isoalloxazine is flanked by two bulky aromatic residues (Tyr and in 3NG7, a Tyr and a Trp). In LkcE, the equivalent residue positions are Gly364 and Leu398, respectively, while in PPOX, the analogous amino acids are M413 and G449. Thus, it appears that the FAD-binding site has been modified in these enzymes in order to accommodate the large macrocycles of the substrates/products.

We would also note that we incorrectly stated that in all structurally-characterized MAO family members, the FAD is covalently bound. In fact, this varies between homologs, and so we have modified the relevant sentence (top of page 4) to reflect this fact: ...; **thus in common with certain MAO family members¹² but distinct from the only other reported amide oxidase Af12070 (ref. ¹⁰), the FAD cofactor is non-covalently bound.**

6. Discussion. Need to add one paragraph discussing that given the highly hydrophobic nature of the substrate binding pocket, how LkcE can be engineered for synthetic biology.

Response:

The referee is correct to point out that this aspect of the active site has implications for the type of substrate which might be processed. We have therefore added the following sentence to page 11:

Although further work will be required to define its substrate profile in detail, these data encourage the idea that LkcE may find wider use as a **ligation/macrocyclization** catalyst in both synthetic biology and organic synthesis applications. **In view of the largely hydrophobic nature of the binding pocket, it will be particularly interesting to explore whether minimally functionalized acyl chains are also substrates.**

7. Fig. 2. Add in SI the mechanism about how compounds 4 to 9 are formed.

Response:

We thank the referee for this suggestion to clarify the origin of certain compounds. In fact, compounds 4 and 5 are commercially-available substrate analogues of LC-KA05 (and thus are not 'formed'). We have added a scheme showing how compounds 6–9 are formed to the front end of Figure S10.

8. Table S3. (1) The R_{free} for wild type is higher than 0.25. Need to explain in SI or add a footnote under Table S3. (2) The B factors of protein and ligand are unusually high. Need to discuss in main text (page 3). (3) Add Ramachandran parameters.

Response:

Concerning the R_{free} for the wild type LkcE, the difference between R_{free} and R_{work} is only 3%, which is of the same order of magnitude as that for the other structures, and the absolute value is only 0.256 (when the R_{free} is expected to be 1/10 of the resolution (so 0.28)). Furthermore, the electron density maps do not show any additional interpretable regions of electron density that would allow for further reducing the R_{work} and R_{free} factors. We therefore, do not think it is necessary to add a footnote to Table S3.

To address the query concerning the B factors, we revisited the Wilson plots calculated from the reduced crystallography data and compared them to those from the refinement (see table below).

	LkcE Se (3.3 Å)	WT (2.8 Å)	R326 (2.5 Å)	E64 (3.03 Å)
Average B- factor on all atoms from the refinement	70.0	87.0	61.0	70.0
B-factor from Wilson plot	75.3	82.2	60.8	72.9

This analysis shows, critically, that the B-factors from the refinement and the Wilson plots are in good agreement. The fact that both are indeed high is not an error, but is rather indicative of significant dynamic disorder at the atomic level. We have added a footnote to this effect to **Supplementary Table 3**.

Concerning the Ramachandran parameters, they are as follows:

LkcE Se: 97.4% (832/854) of all residues were in favored (98%) regions; 99.9% (853/854) of all residues in allowed (>99.8%) regions. 1 outlier (phi, psi): B 313 GLU (55.1, 99.1).

LkcE WT: 96.4% (829/860) of all residues were in favored (98%) regions. 99.7% (857/860) of all residues were in allowed (>99.8%) regions. There were 3 outliers (phi, psi):

A 300 GLY (-15.5, 125.0), B 300 GLY (-15.3, 125.0), B 313 GLU (57.7, 103.9).

E64Q: 96.0% (819/853) of all residues were in favored (98%) regions; 99.8% (851/853) of all residues were in allowed (>99.8%) regions. 2 outliers (phi, psi): A 299 GLY (-57.2, 89.1), B 299 GLY (-58.0, 89.6)

R326Q: 97.4% (829/851) of all residues were in favored (98%) regions; 99.8% (849/851) of all residues were in allowed (>99.8%) regions. 2 outliers (phi, psi): A 300 GLY (-50.0, -103.1), B 300 GLY (52.2, -69.0).

A summary of the requested Ramachandran parameters have been added to the main text (page 4).

For all four structures, >99.7% of the residues were in allowed regions of the Ramachandran plot. In all but the Se LkcE structure one Gly (299 or 300) was an outlier, while Glu313 was an outlier in both forms of the *holo* enzyme structure. However, for Glu313 clear electron density is present corresponding to the residue.

9. All protein figures. Keep the orientation the same when possible to help the readers acclimatize to the structure.

Response:

We have, as requested, maintained the same orientation of LkcE in the figures, except where a change was necessary to visualize a specific aspect of the structure (as in Fig. S3).

10. Fig. 3. (1) Each monomer should be shown in a different color, for example, deep blue vs light blue, deep purple vs light purple. (2) Distances should be shown along the dotted line. (3) Fig 3D, use Pymol to set the slice thinner so that the depth contrast is more obvious.

Response:

We have redrawn the structure of LkcE throughout the ms and SI so that, as requested, each of the monomers is a different color (blue and purple), and the two domains are indicated by different shading (e.g. dark/light blue). Also, for all panels in which there is an active site zoom, we have made the depth contrast more obvious.

11. Fig. 4. Change the view to side by side. Panel A for LkcE dimer, then panel B for hMAO B and panel C for 6HDNO. Then explains in the figure legends that these three were aligned in monomer A.

Response:

As requested we have included three panels in the figure, allowing for a side-by-side comparison of the structures. The legend has also been modified to the following:

Figure 4 | Alternate homodimerization mode of LkcE relative to hMAO B and 6HDNO. (a) Homodimeric structure of LkcE. (b) Homodimerization mode of hMAO B. The dimer is superimposed on one monomer of LkcE (in blue in (a)), in the same orientation. (c) Homodimerization mode of 6HDNO. Again, the dimer has been superimposed on a monomer of LkcE. (d) Interaction surface between the two monomers in LkcE (yellow), hMAO B (red) and 6HDNO (green), relative to a monomer of LkcE. This analysis clearly shows that the mode of homodimerization of LkcE, as well as its overall quaternary organization, differ from these two homologs.

12. In the SI: add one figure with two panels: (1) SA-Omit map of protein active site residues, (2) SA-Omit map of FAD, and (3) SA-Omit map of ligands.

Response:

We do not understand why an omit map has been requested for the protein active site residues, as the sequence of the enzyme is known, and the electron density maps are clear. We have, as requested by the reviewer, added omit maps for the FAD and the ligands to the supplementary information (new **Supplementary Fig. 3**), and renumbered the other figures accordingly.

13. Fig. S2. Make the superposition stereo.

Response:

We would prefer to leave the figure as it is, as it is already quite information-dense, and adding a stereo-element would only in our opinion complicate its interpretation.

14. Fig. S4. Active site residues and dimer interface residues should be marked.

Response:

In fact, these residues were indicated in the original version of the figure (by color-coded boxes), but were obviously not clear enough. We have thus changed the indicators to colored arrows (and see the response to comment 5).

15. Fig. S8. Draw chemical structures (and number them) on the HPLC diagrams to help the readers understand which peaks correspond to which products.

Response:

We thank the reviewer for this useful suggestion. On the other hand, as there is not sufficient space for legible chemical structures, we have elected only to present the structures superimposed on the mass spectra as in the original version, but have added compound numbers or names to the highlighted peaks on the chromatograms to identify them (in white on a black background).

16. Fig. S9. There should be Chemdraw structures of compounds 1 and 7 with 2D NMR correlation marked in arrows.

Response:

As requested, we have added the correlations for compound 1 to Fig. S9 d and f. We were not able to do this for compound 7, however, as we were not able to obtain 2D NMR data on this compound. We did nonetheless prove its structure by difference NMR, as shown in Fig. S9 g.

17. Fig. S10-12. Label the compound numbers in color on the HPLC peaks.

Response:

This has been done for all figures, and see the response to query 15.

18. Fig. S13. Suggest changing to double reciprocal. As is, it looks bad without the pre-saturation data points. Also, it should be discussed that without the pre-K_m points, the kinetic curve may miss the sigmoidal curve, an indication of allosteric interactions between two monomers.

Response:

We would prefer not to present the data in double reciprocal form, as transformation of kinetic data amplifies any errors, thus giving poorer estimates of the kinetic parameters (Enderle (2012) *Introduction to Biochemical Engineering*, section 8.2.2). In response to the second comment, we have added the following explanation to the legend to the figure:

It must be noted that as we were limited in these assays for reasons of sensitivity to higher concentrations of substrate, it is possible that we missed an earlier sigmoidal dependence on concentration, indicative of cooperative behavior between the two LkcE monomers.

Reviewer #2 (Remarks to the Author):

The manuscript by Weissman and colleagues describes the structural and functional analysis of LkcE, a monoamine oxidase enzyme involved in the maturation of the hybrid PKS/NRPS natural product lankacidin. The enzyme catalyzes the oxidation of an amide, followed by the subsequent carbon-carbon bond forming step characterized as an intramolecular Mannich reaction. The structure of the native protein is determined by single wavelength anomalous methods, characterized, and compared to other MAO family members. The substrate for LkcE is isolated from a mutant strain, characterized, and used in both kinetic and structural studies, supporting a structural mechanism that involves a significant conformational change in the enzyme to enable the intramolecular cyclization step to occur. This is an interesting enzyme that exists within an equally interesting biosynthetic pathway.

There are a few minor issues that preclude complete evaluation at this stage. I also have some suggestions to make some of the data more accessible.

1. The structural data are sound. The structures, while moderate resolution, show good statistics that suggest the structural conclusions are warranted. The authors provide preliminary validation reports that also do not raise any concerns. However, the preliminary validation reports are not completed for the LckE substrate ligand, molecule DRG. More important, no electron density is provided for any of the ligands in Figure 3b-d. For a large, flexible ligand such as used here, it is common for the density to be less than ideal. Particularly at the resolution of the current structures, the authors should provide experimental evidence for the structural conclusions concerning the ligands in the main text.

Response:

For the first comment of the reviewer concerning molecule DRG, it has been renamed as CWH by the curators of the PDB, explaining why the report appeared incomplete.

In response to his/her second comment, we have now added electron density to the ligands in Fig. 3b–d, and furthermore, the corresponding omit maps have been provided as an additional figure (**Supplementary Fig. 3** (and see response to Reviewer 1, comment 12)). As a consequence of the resolution of the data sets (ca. 3 Å) and the high B factors from the Wilson plots, the refinements were carried out using Buster in order to improve the quality of the electron density maps. Compared to other commonly-used refinement programs, Buster is particularly well adapted by implementing numerous cycles of model construction and omit map calculations, to improving the phases and identifying and constructing without ambiguity flexible loop regions and ligands. Globally, this gave good fits between the ligands and the electron density (see **Fig. 3** and **3S**), although less well to the most flexible part of the ligand EMEA.

2. Assuming it does not conflict with figure/table limits of the journal, the mutagenesis data should be moved to the main text. These are some of the more significant results of the paper and they should not be relegated to the SI. (The current figure 4, showing the oligomerization and interface, is less important than the functional data and could be moved to the SI). Regardless of whether they are kept in the SI or moved to the text, the five initial velocity plots should be presented on a single page, allowing better comparison. Finally, I would prefer to see the kinetic data also presented in a table in addition to just the text.

Response:

We would prefer to keep Fig. 4 in the main text, and the kinetic data in the SI (though all of the values are in fact presented in the main text, though not in tabular form). As requested by the reviewer, we have created a single figure showing the kinetic data (Fig. S13), and summarized the values in an accompanying table. We have also modified the K_M values of the LckE mutants to a single significant figure to match those of the wild type.

3. The authors should also consider whether it might be more effective to divide the LC-MS data of the SI into separate figures. This would allow the relevant legend to be presented on each page with the panels that describe each experiment. Figure S8 could be divided with panels a-e, f-h, i-m, n-r, and s-w as currently organized but with the legend present on the same page as each. The same could be done for Figures S10 and S12. In figure S12, each set of UV/LCMS-XID/and MS spectra should also contain a panel (perhaps above each series of panels) showing the chemical reaction for what is being performed. For example, on page s35 with panels a-e, the top of the page should show: the substrates, the inactivated enzyme. The page s36 with panels f-j should show the same substrates, the active enzyme, and the product Lankacidinol A (2). As with kinetics above, this is critical for the manuscript and the authors may wish to consider whether an extracted summary of these data can form a figure within the main body of the manuscript.

Response:

We thank the reviewer for helpful suggestions on improving our data presentation. As requested, we have grouped the legends of Figs. S8, and S10–S12 with their appropriate chromatograms/spectra. We have also added reaction schemes to Fig. S12 illustrating each of the *in vitro* experiments. **As for whether a portion of these data should be presented as an additional figure in the main text, we ask the journal for advice.**

4. The authors propose an interesting evolutionary pathway toward the new activity involving a large conformational change. While possible, I am not sure that this is sufficiently supported to allow statements like “Our results show that the new catalytic activity depends intimately on a change in the mode of dimerization of the enzyme relative to classical MAO family...” (pg 10). This may be true; however, there may be many features that are required for this new activity.

Response:

We agree that we perhaps overstated this conclusion. We have thus modified the relevant sentence (page 11) to read: Our results **strongly suggest** that the new catalytic activity **of LkcE depends, among other factors**, on a change in the mode of dimerization of the enzyme relative to classical members of the MAO family...